# Nourishing the Brain or the Mood? Dietary Omega-3s for Psychological, but Not Cognitive Health

**DOI:** 10.3390/nu18010050

**Published:** 2025-12-23

**Authors:** Jakub Orłowski, Maria Kossowska-Wywiał, Aneta Brzezicka

**Affiliations:** Institute of Psychology, SWPS University, 03-815 Warsaw, Poland; jorlowski@swps.edu.pl (J.O.); mkossowska1@swps.edu.pl (M.K.-W.)

**Keywords:** omega-3 fatty acids, dietary patterns, depression, stress, memory, cognition, healthy adults

## Abstract

**Background**: Mood disturbances, often accompanied by cognitive deficits, represent a major public health challenge. Diet is increasingly recognized as a modifiable factor in mental health, with specific nutrients such as omega-3 polyunsaturated fatty acids (PUFAs) showing therapeutic potential. This study investigated whether dietary omega-3 intake moderates variations in cognitive performance associated with psychological symptoms in non-clinical samples. **Method**: A total of 313 healthy adults completed a food frequency questionnaire (FFQ) to assess dietary intake. Psychological symptoms were measured using the depression screening questionnaire (PHQ-9) and the stress assessment instrument (PSS-10), while cognitive performance, including short-term and episodic memory, was evaluated using Sternberg and Old/New recognition tests. A subgroup of 52 older adults completed a detailed in-person FFQ to enable the precise quantification of EPA, DHA, and alpha-linolenic acid (ALA) intake. **Results**: Diets rich in omega 3, particularly nuts, seeds, fish and seafood, were associated with lower depression and stress scores. EPA and DHA, but not ALA, were specifically linked to those mood benefits. However, dietary omega-3 intake was not significantly associated with cognitive performance and did not moderate the relationship between mood and memory. Self-reported omega-3 supplementation was linked to fewer depressive symptoms and better episodic memory. **Conclusions**: While dietary omega-3 is associated with improved well-being, its role in cognition in healthy adults is not well-established and warrants further investigation.

## 1. Introduction

Mood disorders, often co-occurring with cognitive impairments, represent a significant global health challenge. Affecting an estimated 300 million individuals, Major Depressive Disorder (MDD) stands as one of the most common mental illnesses globally, with projections indicating it will become the foremost cause of disease burden by the year 2030 [1]. In Poland, epidemiological surveys estimate that around 3–4% of adults meet the criteria for depressive episodes and around 10% of its subclinical form. Given the pervasiveness of subclinical symptoms in the general population, early mood disturbances can significantly impair quality of life and serve as a clear marker of elevated risk for developing a full-blown disorder, such as MDD [2]. Consequently, the search for early and preventive interventions to mitigate subclinical symptoms is a major priority in clinical research.

The pathophysiology of mood disorders is complex and involves several mechanisms, including dysregulation of the hypothalamic–pituitary–adrenal (HPA) axis, neurotransmitter systems (e.g., serotonin, dopamine), neuroinflammation, and structural and functional brain changes in regions such as the hippocampus, prefrontal cortex, and amygdala [3]. These neurological systems are also influenced by nutritional status [4]. Dietary patterns, particularly the Western diet pattern, high in saturated fats and refined sugars, have been associated with an increased risk of depression, which is likely mediated through pro-inflammatory pathways of cytokine-driven neuroinflammation (e.g., IL-6, TNF-α, IL-1β), activation of the kynurenine pathway via the indoleamine 2,3-dioxygenase enzyme, and systemic inflammation driven by lipopolysaccharide translocation resulting from increased intestinal permeability [5]. Conversely, anti-inflammatory diets, such as the Mediterranean diet, which is rich in fruits, vegetables, fish, and abundant in healthy fats like omega-3 fatty acids, demonstrate preventive and even mood-improving effects [6,7]. Omega-3 polyunsaturated fatty acids (PUFA), specifically their long-chain forms such as eicosapentaenoic acid (EPA) and docosahexaenoic acid (DHA) from marine sources, as well as alpha-linolenic acid (ALA) from plants, are essential nutrients recognized for their potent neuroprotective and anti-inflammatory properties [8,9]. Their roles in the brain are distinct yet complementary. DHA serves as a fundamental structural component of neuronal membranes, crucial for brain development, synaptic function, and plasticity [10], while EPA is primarily involved in modulating anti-inflammatory processes and neurotransmitter function, which are fundamental for regulating mood [11]. Complementing these specialized roles, ALA functions primarily as a potent, universal antioxidant and mitochondrial co-factor, providing fundamental neuroprotection by mitigating oxidative stress [12]. Despite compelling biological synergy, the clinical evidence for omega-3 fatty acids remains mixed, especially when transitioning from treatment efficacy to primary prevention.

Most research has focused on clinical populations or pharmacologic doses of supplements [13,14]. Studies examining the relationship between habitual dietary intake of omega-3s (a sustainable and public-health metric) and subclinical mental health in the general adult population yield conflicting results [15,16]. Some longitudinal studies in broad adult populations suggest a protective effect of dietary omega-3 against the incidence and persistence of depressive episodes [17], while others, often limited to specific subgroups, have found no significant association [18]. Furthermore, while diets rich in omega-3s are associated with a long-term reduced risk of cognitive decline [19,20] it is unclear whether this protective effect extends to moderating the subtle cognitive costs of poor mood in healthy individuals. Therefore, we hypothesize that a higher habitual dietary intake of omega-3 PUFAs will moderate the negative relationship between mood symptoms and cognitive performance in the non-clinical adult population.

## 2. Materials and Methods

### 2.1. Data Collection and Study Protocol

The study utilized a two-phase sequential design. Participants were recruited throughout 2023 and 2024 via a multi-channel approach, including social media advertisements, lectures, and presentations organized at local nursing homes and senior centers in Warsaw, Poland. This outreach was supplemented by the distribution of flyers and posters. Inclusion criteria for initial participation required individuals to be over 19 years of age, have no self-reported history of neurological disorders, and reside within the Warsaw metropolitan area.

The first phase of the study involved conducting an online cross-sectional survey designed to assess dietary patterns, psychological well-being, and cognitive performance. Prior to participation, individuals were informed regarding the study’s aims and provided informed consent. Demographic and wellness data were collected using the Qualtrics platform [21]. Cognitive tasks were developed in PsychoPy application [22] and hosted via Pavlovia.org. The entire procedure was completed remotely through a single online link. This approach was chosen for feasibility, data integrity, and standardization of experimental conditions. However, it may have inadvertently restricted participation among individuals with limited Internet access or lower digital literacy, thereby leading to an overrepresentation of some subgroups, e.g., participants with higher education levels.

For the second phase of the study, a sub-cohort of individuals aged 50 years or older was selected from the initial online pool and invited for in-person assessment at the NeuroCognitive Research Center in Warsaw, Poland. Upon arrival, participants provided written informed consent for the follow-up session. The Mini-Mental State Examination (MMSE; [23]) was administered as a screening tool for global cognitive health. An eligibility cut-off score of ≥28 points was set to ensure the inclusion of cognitively intact individuals. Participants who met this criterion then completed an expanded, detailed dietary questionnaire, which featured a greater number of items than the online survey, to obtain more reliable dietary data.

### 2.2. Dietary Assessment and Omega-3 Daily Intake

Dietary intake was evaluated using a modified 83-item Polish Food Frequency Questionnaire (FFQ; [24]) referencing the previous 12 months. For each listed food item or composite dish, participants selected their consumption frequency from a six-point scale (“never”, “1–3 times per month”, “once per week”, “several times per week”, “once per day”, and “several times per day”). For this study, an Omega-3 Diet Quality Index (omega-3 DQI) was specifically developed, adapting the framework of the Pro and Non-Healthy-Diet-Index-10 from the KomPAN manual [25], to provide a quantitative measure of omega-3-rich food consumption. The index incorporated eight dietary sources, which are main sources of omega-3 polyunsaturated fatty acids [26], identified from the FFQ options: “Fish and seafood (e.g., smoked fish, marinated fish, fish in oil, cream, canned fish, fried, boiled, seafood)”, “Fatty fish (e.g., Salmon, Tuna, Cod liver oil)”, “Oils (e.g., coconut, sunflower, rapeseed)”, “Fatty oils (linseed, rapeseed, soybean)”, “Nuts (e.g., various nuts, peanut butter, chocolate-nut spread)”, “Walnuts”, “Chia seeds”, “Flaxseeds”. Consumption frequencies were converted to daily equivalents using established conversion factors (never = 0; 1–3 times per month = 0.06; once per week = 0.14; several times per week = 0.5; once per day = 1; several times per day = 2). The omega-3 DQI was calculated asomega-3 DQI = 100 × (sum of daily frequencies for 8 selected items)/16,
generating scores from 0 to 100 with higher values reflecting a greater consumption of omega-3 rich foods.

A subsample from the second phase of the study completed a detailed, 405-item FFQ, administered through in-person, structured interviews to assess habitual dietary intake over the preceding month. Portion sizes were quantified using visual aids, including illustrations, household measures, hand-size guides, and original product packaging [27,28]. The average daily intake (g/day) of ALA, EPA and DHA was calculated based on the reported frequency and quantity of chosen food consumption. The assessed items encompassed fish and seafood, including both lean and fatty species (e.g., cod, salmon, mackerel, trout, tuna, herring); fish products (e.g., smoked, salted, and canned varieties such as sardines and herring in various sauces); fats and oils (e.g., linseed, rapeseed, soybean, and olive oils, as well as margarine, butter, and lard); nuts and seeds (e.g., walnuts, almonds, hazelnuts, and pumpkin, sunflower, and sesame seeds). For each food item, its omega-3 fatty acid content per 100 g was obtained from food composition tables. This value was then multiplied by the reported consumption frequency (converted to a daily equivalent) and the standard portion size (in grams) to estimate the daily contribution from each food. The individual values were subsequently summed to derive the total estimated daily intake for each fatty acid. These calculations were based on the Polish National Food and Nutrition Institute food-composition tables [29], with missing values supplemented with data from the USDA National Nutrient Database for Standard Reference.

### 2.3. Psychological Well-Being Questionnaire

Depressive symptoms were assessed using the Polish version of the Patient Health Questionnaire-9 (PHQ-9; [30]). The instrument comprises nine items, each rated on a 4-point Likert scale ranging from 0 to 3. The total score, which is the sum of all item scores, ranges from 0 to 27, with higher scores indicating greater severity of depressive symptoms. Cronbach’s alpha for this instrument in the current study was α = 0.84.

Perceived stress was evaluated using the Polish adaptation of the Perceived Stress Scale-10 (PSS-10; [31]). This 10-item scale employs a 5-point Likert scale (0–4). The total score, derived from the sum of all items (including four reverse-scored items), ranges from 0 to 40, with higher scores reflecting higher levels of perceived stress. Cronbach’s alpha for the instrument in the current study was α = 0.88.

### 2.4. Short-Term and Episodic Memory Assessment

Both memory tasks were objective, performance-based computerized tasks; no cognitive outcomes were self-reported. Episodic memory was evaluated using an old/new recognition task consisting of learning and recognition phases. During the learning phase, participants were presented with 30 images representing healthy and unhealthy foods, healthy and unhealthy dishes, and clothing items. To ensure sustained attention, they were prompted eight times at random intervals to indicate whether the preceding image was an edible item. After a 10 min break occupied with a short-term memory task, participants proceeded to the recognition phase, in which 60 images (30 “old” and 30 “new”) were presented in a randomized sequence. Each image was displayed for one second, followed by a response window of 2.3 to 2.8 s for participants to indicate whether the image had appeared in the learning phase. Memory performance was quantified using signal detection theory [32], with sensitivity (d’) calculated asd’ = z(H) − z(F),
where z(H) represents the z-score of the Hit Rate (H), and z(F) the z-score of the False Alarm Rate (F).

To assess short-term memory, participants completed a task adapted from the Sternberg paradigm. Participants were presented with memory sets containing between 2 and 7 abstract symbols, each displayed for 1 s with a 200 ms inter-stimulus interval. Following a 1 s delay after the last symbol, a single probe symbol was presented, and participants were required to indicate whether it had been present in the preceding memory set or not. Performance was analyzed to calculate working memory capacity (K) for each set size, following Vogel’s data analysis [33], according to the formulaK = S × (H − F),
where S represents the set size. Based on the K values obtained across different set sizes, overall working memory capacity was estimated by identifying the largest set size for which participants consistently maintained accurate performance.

### 2.5. Statistical Analysis

All data were analyzed using SPSS software (version 29). Principal Component Analysis (PCA) with Varimax rotation was applied to the food frequency data to identify the underlying consumption frequency of omega-3 rich foods, and the adequacy of the sample was assessed using the Kaiser–Meyer–Olkin (KMO) test. To assess the relationships between the various dietary omega-3 intake measures and the outcome variables, Spearman’s rank correlation was used. A linear regression analysis was then employed to evaluate the relationship of psychological well-being on memory scores. Subsequently, a moderation analysis was conducted via Hayes’ PROCESS macro (Model 1) to determine whether omega-3 intake moderated the aforementioned relationship. The bootstrap method with 5000 samples and 95% confidence intervals was applied to verify the effects. Finally, group differences in psychological and memory scores based on supplement usage were assessed with the Mann–Whitney U test. A value of *p* < 0.05 was considered statistically significant for all tests.

## 3. Results

### 3.1. Characteristics of Participants

The total sample included 313 participants, mostly female (87.5%) with a mean age of 40.3 years (SD 15.29) (Table 1). The majority were employed (72.5%) and highly educated (78.6%), with a mean BMI of 23.99 (SD 4.19). Few participants reported smoking (9.9%), and assessing physical activity, levels were low in nearly half the sample (49.2%).

The subsample from the second phase (N = 52) was similar to the main groups with 88.5% female, mean age 58.3 years (SD 6.02), mostly employed (78.8%), and highly educated (86.5%) with mean BMI of 26.85 (SD 6.61). Smoking was negligible (1.9%), and physical activity levels were comparable.

The below tested statistical models’ structure was theory-driven and not compared against alternative model specifications.

### 3.2. Omega 3 Dietary Habits

First, dietary categories from online data of foods rich in omega-3 (fish and seafood, fatty fish, fatty oils, nuts, walnuts, chia seeds and flaxseeds) were identified using Principal Component Analysis (PCA). The suitability of the data for factor analysis was confirmed by a significant Bartlett’s test of sphericity (*χ^2^*(28) = 609.18, *p* < 0.001) and a Kaiser–Meyer–Olkin (KMO) measure of 0.63. The analysis yielded a three-component solution, which explained a total of 67.23% of the variance in the consumption of omega-3 rich foods. Following rotation, Component 1 accounted for 29.85% of the variance, Component 2 for 21.16%, and Component 3 for 16.23%. The components revealed clear dietary categories, which were then labeled as (1) Nuts and Seeds, (2) Fish and Seafood, and (3) Oils, based on the items with the highest factor loadings (>0.70) (Table 2).

### 3.3. Associations Between Omega-3 Intake and Mood

We examined associations between omega-3 dietary habits intake and mood scores, with significant negative correlations for omega-3 intake measures. For the specific questionnaires, significant negative correlations were obtained with the Omega-3 DQI (PHQ-9: *ρ* = −0.226, *p* < 0.001; PSS-10: *ρ* = −0.172, *p* = 0.002), the Nuts and Seeds component (PHQ-9: *ρ* = −0.223, *p* < 0.001; PSS-10: *ρ* = −0.164, *p* = 0.004), and the Fish and Seafood component (PHQ-9: *ρ* = −0.192, *p* < 0.001; PSS-10: *ρ* = −0.157, *p* = 0.005) (Table 3). Analysis of the estimated fatty acid intake in the subsample revealed significant negative correlations for EPA (PHQ-9: *ρ* = −0.362, *p* = 0.008; PSS-10: *ρ* = −0.314, *p* = 0.023) and DHA (PHQ-9: *ρ* = −0.325, *p* = 0.019; PSS-10: *ρ* = −0.327, *p* = 0.018). No statistically significant correlations were observed between the intake of Oils component or ALA and PHQ-9 and PSS-10 scores (*p* > 0.05).

### 3.4. Associations Between Omega-3 Intake and Memory

To examine the relationship between various aspects of omega-3 dietary intake habits and short-term and episodic memory performance, correlation analysis was conducted. All tested variables showed weak correlations with the memory indices, and none of them were statistically significant (*p* > 0.05) (Table 4).

### 3.5. Relationship Between Mood and Cognition, Check Before Moderation

Before conducting the moderation analysis, correlations were performed between the predictor variables (PHQ-9 and PSS-10) and the outcome measures (d’ and K), to assess the strength and direction of the relationships. A significant negative correlation was found between PHQ-9 scores and long-term memory performance (d’) (*ρ* = −0.118, *p* = 0.037) (Table 5). No correlation was found between PSS-10 and d’ or between mood scores and working memory (K).

A linear regression analysis was conducted to examine whether the severity of subclinical depression, measured with the PHQ-9 questionnaire, is a predictor of long-term memory performance. The regression model with subclinical depression as a predictor was significant (*R^2^* = 0.018, *F*(1, 311) = 5.664, *p* = 0.018), explaining 1.8% of the variance in long-term memory scores (Table 6). The regression coefficient for the PHQ-9 scale was negative (*B* = −0.019, *t* = −2.380, *p* = 0.018).

### 3.6. Omega-3, Memory and Mood, Moderation Model

The model did not explain a significant proportion of the variance in the dependent variable d’ (*R^2^* = 0.023, *F*(3, 309) = 2.351, *p* = 0.072) (Table 7). Among the predictors, PHQ-9 score emerged as a significant predictor (*b* = −0.028, *t*(309) = −2.031, *p* = 0.043), whereas omega-3 DQI was not significant. The interaction between depression and diet quality was also non-significant. Furthermore, the highest-order interaction between PHQ-9 and omega-3 DQI did not significantly predict the dependent variable (Δ*R*^2^ = 0.003, *F*(1, 309) = 0.931, *p* = 0.336).

The moderation analyses did not provide evidence that habitual omega-3 intake alters the relationship between depressive symptoms and episodic memory. While depressive symptoms showed a small but significant negative association with memory performance, dietary omega-3 intake did not interact with this effect. Given the absence of moderation, subsequent analyses focused on other aspects of omega-3 status, including differences between individuals who reported taking omega-3 supplements and those who did not.

### 3.7. Differences Between Supplement Users and Non-Users

Fifty-two participants (16.6%) reported using omega-3 supplements, based on self-reported descriptions of supplement use, which included fish oil, krill oil, cod liver oil, DHA oil, or pure fish oil. Supplement users had significantly lower PHQ-9 scores (U = 5520.00, Z = −2.13, *p* = 0.033) and significantly higher d′ scores (U = 5585.50, Z = −2.02, *p* = 0.044) compared to non-users (Table 8). The difference in PSS-10 scores approached significance (U = 5628.50, Z = −1.95, *p* = 0.052), with lower scores observed in supplement users. No significant difference was found in short-term memory capacity (K) between the groups.

Overall, higher dietary omega-3 intake was associated with better mood but not with memory performance, and the moderation analyses did not support an interaction between mood and diet. Notably, supplement users showed fewer depressive symptoms and better episodic memory than non-users, suggesting that more consistent omega-3 intake may be relevant for psychological well-being and cognition.

## 4. Discussion

This study investigated the associations between dietary omega-3 intake, psychological well-being, and cognitive function in healthy adults. While our initial hypothesis that omega-3 intake would moderate the relationship between mood and memory was not supported, the results robustly indicate a significant association between the consumption of products rich in omega-3, particularly nuts, seeds, fish, and seafood, and the severity of depressive symptoms and stress. This observation is consistent with studies indicating the protective or beneficial role of fish [34], nuts [35], and seeds [36] in the context of depression risk. The strongest associations with individual fatty acids were observed for the long-chain EPA and DHA, confirming their primary importance in mood regulation, likely due to their roles in modulating neurotransmission and the immune response in the central nervous system [37]. In contrast, the lack of a significant association between ALA and mood disorder symptoms may result from its low and inefficient conversion to the biologically active forms, EPA and DHA, in the human body [38].

In contrast to the clear benefits for mood, no significant association was found between dietary omega-3 intake and the efficiency of either episodic or short-term memory. Furthermore, the conducted moderation analysis ruled out the possibility that a diet rich in omega-3 products mitigates the negative impact of depressive symptoms on memory. Most cognitive benefits are often observed in populations at elevated risk of deficits, such as the elderly or individuals with mild cognitive impairment [39]. Consequently, our findings, while reinforcing the link with mood, do not provide a basis for immediate clinical changes or dietary recommendations aimed at cognitive enhancement or mood regulation in generally healthy adults. Current evidence remains exploratory rather than directive. In a healthy adult population, whose cognitive functions are within the normal range, an increased supply of these fatty acids may not exert any further, measurable effect. Conversely, a direct negative impact of the severity of depressive symptoms (PHQ-9) on episodic memory was observed, which is consistent with the literature documenting deficits in the recall of episodic details in a dysphoric mood [40]. The absence of a link between stress and memory could be because general self-reported stress is too broad to reliably affect performance on a specific memory task. The effect can be concealed by individual differences in coping strategies, making a consistent pattern difficult to observe [41].

The fact that individuals reporting omega-3 supplementation had better results in episodic memory tests and a reduced number of depressive symptoms compared to those not taking supplements is particularly meaningful. Unlike dietary intake, which varies substantially in fatty acid composition depending on food type, preparation method, and portion size, supplementation provides a more stable and standardized dose of EPA and DHA. These long-chain fatty acids also have higher bioavailability in supplemental form, allowing individuals to reach physiologically relevant concentrations more consistently than through diet alone. This suggests that to obtain a pro-cognitive effect in healthy individuals, it may be necessary to achieve a higher, more standardized, and consistent intake level of EPA and DHA than is typically possible through a conventional diet alone [42]. Additionally, long-term, high doses of daily use omega-3 supplements, not just as a short-term intervention, may yield better results and be associated with longer preservation of cognitive function in healthy adults [43]. This difference highlights the discrepancy between interventional potential (supplementation) and observational effects (diet), underscoring the importance of dose optimization and population-specific context in future cognitive and neuropsychiatric research. However, the cross-sectional nature of this study precludes causal inferences and complicates the interpretation of the observed directionality. It is equally plausible that individuals in a dysphoric mood adopt poorer dietary habits, consequently reducing their intake of micronutrients important for mental health [44]. This may create a cycle in which a worse mood is both a cause and a consequence of low omega-3 intake, rather than merely its result. To definitively disentangle these effects and establish causality, future research must prioritize longitudinal designs that track dietary habits and mental health outcomes over extended periods.

The structure of the sample in this study was skewed towards women and highly educated individuals, which may limit the generalizability of the findings [45]. This is likely attributable to a volunteer bias, as both demographic groups typically demonstrate a greater propensity to participate in health-related surveys [46]. Although onsite recruitment was conducted at locations like senior centers to diversify participation, this volunteer bias was likely exacerbated by the primary reliance on online recruitment and data collection methods for the initial phase, which may have been less accessible to individuals with lower socioeconomic status or digital literacy. It is also important to consider that the effects of nutritional omega-3 fatty acids may differ across specific populations. Young people, in particular, represent a fragile subgroup that is both vulnerable to mood disturbances and more susceptible to adverse events associated with pharmacological treatments [47]. Likewise, older adults constitute another sensitive population in which dietary omega-3 intake may have distinct implications, given the increased risk of cognitive decline, neuroinflammation, and comorbidities that can influence both mood regulation and nutrient metabolism.

The subjective assessment of omega-3 fatty acid intake, which relies on self-report online questionnaires, is a notable limitation [48] due to the difficulty of remembering which foods they consumed, in what quantities, and how frequently, potentially leading to an underestimation or overestimation of actual intake. Crucially, the lack of verification through objective biomarkers, such as plasma levels of EPA and DHA, means that the actual biological exposure remains unconfirmed. Overcoming this limitation in future work through the inclusion of biomarker assays is essential to move from associative dietary data to a more direct understanding of nutrient status and its effects. Furthermore, the use of broad food categories, without separating specific products or dishes, could introduce measurement error, as preparation methods (e.g., baked, fried, processed) that significantly affect omega-3 content were not considered [49]. Future studies should incorporate objective biomarkers and collect more granular data on food preparation to improve the accuracy of dietary intake estimates. Another important limitation is that we did not have direct control over the participants’ actual food intake. Self-reported dietary data are vulnerable to recall biases and may not fully capture qualitative aspects of eating behavior. It cannot be ruled out that the observed associations are confounded by an overall healthier diet or other unaccounted lifestyle factors. Although the results for mood symptoms are consistent with existing literature, their assessment was subjective. To fully understand the impact of omega-3 fatty acids on mood disorders and the brain, studies utilizing neuroimaging (fMRI) or electrophysiology (EEG) are essential. These methods could reveal subtle changes in brain function or structure that are not detectable through behavioral cognitive tests and help clarify the pathway through which omega-3 influences mood and cognition [50]. Finally, while this study focused on diet as a moderator, future research should consider alternative models, including the role of diet as a mediator. It is plausible that omega-3s (particularly from supplementation) primarily affect mood, and this mood improvement, in turn, mediates subsequent changes in cognitive performance. In summary, this observational study adds to the foundational evidence on omega-3s and mental well-being while highlighting the methodological steps, like longitudinal designs, biomarker validation, and mechanistic exploration, required to assess their potential for clinical or public health application.

## 5. Conclusions

This study provides evidence that habitual dietary intake of omega-3 fatty acids, particularly from marine sources, is associated with reduced psychological distress in a non-clinical adult population. However, such dietary intake was not associated with memory performance and did not moderate the link between mood and cognition. The more pronounced benefits observed in omega-3 supplement users for both mood and episodic memory suggest that it can be key to supporting cognitive health. These findings underscore the importance of distinguishing between dietary and supplemental sources of omega-3s and highlight the need for future research that uses objective biomarkers, diverse samples, and longitudinal designs to clarify their distinct roles in promoting psychological and cognitive well-being.

## Figures and Tables

**Table 1 nutrients-18-00050-t001:** Summary of participant characteristics and descriptive statistics.

Variable		N (%)	N (%)
		Phase 1 (N = 313)	Phase 2 (N = 52)
Sex			
	Female	274 (87.5)	46 (88.5)
	Male	39 (12.5)	6 (11.5)
Age (years)			
	Minimum	20	50
	Maximum	90	73
	Mean	40.30	58.25
	SD	15.29	6.02
Employment			
	Employed	227 (72.5)	41 (78.8)
	Unemployed	86 (27.5)	11 (21.2)
Education			
	Primary	0 (0)	0 (0)
	Secondary	67 (21.4)	7 (13.5)
	Higher/University	246 (78.6)	45 (86.5)
Body Mass Index (kg/m^2^)			
	Minimum	13.84	17.72
	Maximum	39.18	39.18
	Mean	23.76	26.13
	SD	4.19	4.41
Smoking			
	Yes	31 (9.9)	1 (1.9)
	No	282 (90.1)	51 (98.1)
Physical Activity			
	Low	154 (49.2)	24 (46.2)
	Moderate	140 (44.7)	23 (44.2)
	High	19 (6.1)	5 (9.6)

SD: standard deviation.

**Table 2 nutrients-18-00050-t002:** Factor loadings of omega-3-Rich Food Items across the three extracted dietary groups: Nuts and Seeds, Fish and Seafood, and Oils.

FFQ Selected Items	Nuts and Seeds	Fish and Seafood	Oils
Nuts (e.g., various nuts, peanut butter, chocolate-nut spread)	0.698	0.083	0.178
Walnuts	0.792	0.157	0.183
Flaxseeds	0.812	−0.056	−0.003
Chia seeds	0.762	0.052	−0.092
Fish and seafood (e.g., smoked fish, marinated fish, fish in oil, cream, canned fish, fried, boiled, seafood)	0.000	0.913	0.084
Fatty fish (e.g., Salmon, Tuna, Cod liver oil)	0.152	0.89	−0.079
Oils (e.g., coconut, sunflower, rapeseed)	0.074	−0.125	0.796
Fatty oils (linseed, rapeseed, soybean)	0.062	0.125	0.76

FFQ = Food Frequency Questionnaire.

**Table 3 nutrients-18-00050-t003:** Correlations between dietary intake measures and mood scores.

Measure	PHQ-9 *ρ*	PHQ-9 *p*	PSS-10 *ρ*	PSS-10 *p*
Dietary Omega-3 Sources and Indices				
Omega-3 DQI	−0.226	<0.001	−0.172	0.002
Nuts and Seeds	−0.223	<0.001	−0.164	0.004
Fish and Seafood	−0.192	<0.001	−0.157	0.005
Oils	−0.007	0.908	−0.020	0.732
Fatty Acid Intake (Phase 2)				
ALA	0.194	0.168	0.063	0.656
EPA	−0.362	0.008	−0.314	0.023
DHA	−0.325	0.019	−0.327	0.018

Statistical analysis involved Spearman’s rank-order correlation to assess the associations between variables. The *ρ* (rho) coefficient is reported. PHQ-9 = Patient Health Questionnaire-9; PSS-10 = Perceived Stress Scale-10; Omega-3 DQI = Omega-3 Diet Quality Index; ALA = Alpha-Linolenic Acid; EPA = Eicosapentaenoic Acid; DHA = Docosahexaenoic Acid.

**Table 4 nutrients-18-00050-t004:** Correlations between dietary intake measures and cognitive scores.

Measure	d’ *ρ*	d’ *p*	K *ρ*	K *p*
Dietary Omega-3 Sources and Indices				
Omega-3 DQI	0.071	0.212	0.023	0.687
Nuts and Seeds	0.065	0.260	−0.032	0.577
Fish and Seafood	−0.046	0.426	−0.065	0.258
Oils	−0.059	0.311	−0.058	0.320
Fatty Acid Intake (Phase 2)				
ALA	−0.029	0.837	−0.048	0.736
EPA	−0.026	0.856	0.024	0.865
DHA	−0.112	0.429	0.059	0.676

Statistical analysis involved Spearman’s rank-order correlation to assess the associations between variables. The *ρ* (rho) coefficient is reported. d’ = sensitivity index in the episodic memory task; K = capacity estimate in the short-term memory task; Omega-3 DQI = Omega-3 Dietary Quality Index; ALA = Alpha-Linolenic Acid; EPA = Eicosapentaenoic Acid; DHA = Docosahexaenoic Acid.

**Table 5 nutrients-18-00050-t005:** Correlations between psychological and memory measures.

Measure	d’ *ρ*	d’ *p*	K *ρ*	K *p*
PHQ-9	−0.118	0.037	−0.014	0.801
PSS-10	−0.061	0.283	0.003	0.963

Statistical analysis involved Spearman’s rank-order correlation to assess the associations between variables. The *ρ* (rho) coefficient is reported. PHQ-9 = Patient Health Questionnaire-9; PSS-10 = Perceived Stress Scale-10; d’ = sensitivity index in the episodic memory task; K = capacity estimate in the short-term memory task.

**Table 6 nutrients-18-00050-t006:** Linear regression predicting long-term memory performance from subclinical depression.

Predictor	*B*	SE	*β*	*t*	*p*
Constant	3.256	0.068		47.778	<0.001
PHQ-9	−0.019	0.008	−0.134	−2.380	0.018

Results of a linear regression analysis. *B* = unstandardized regression coefficient; SE = standard error; *β* = standardized regression coefficient. Dependent variable: d’. PHQ-9 = Patient Health Questionnaire-9.

**Table 7 nutrients-18-00050-t007:** Multiple regression of episodic memory performance on depression, omega-3 indices, and their interaction.

Predictor	*b*	SE	*t*	*p*	Lower 95% CI	Upper 95% CI
(Constant)	3.278	0.126	26.061	<0.001	3.030	3.525
PHQ-9	−0.028	0.014	−2.030	0.043	−0.056	−0.001
Omega-3 DQI	−0.003	0.009	−0.314	0.753	−0.020	0.014
PHQ-9 × Omega-3 DQI	0.001	0.001	0.965	0.335	−0.001	0.003

Multiple linear regression analysis testing for moderation (Model 1). *b* = unstandardized regression coefficient; SE = standard error; CI = confidence interval. Omega-3 DQI = Omega-3 Diet Quality Index, PHQ-9 = Patient Health Questionnaire-9.

**Table 8 nutrients-18-00050-t008:** Comparison of scores between groups with and without omega-3 supplementation.

Measure	Omega 3 Supplement Users (Mean Rank)	Omega 3 Non-Users (Mean Rank)	*Z*	*p*
PHQ-9	132.65	161.85	−2.129	0.033
PSS-10	134.74	161.43	−1.945	0.052
d’	180.09	152.40	−2.017	0.044
K	158.63	156.67	−0.146	0.884

Group comparisons using the Mann–Whitney U test; Z-scores are reported. PHQ-9 = Patient Health Questionnaire-9; PSS-10 = Perceived Stress Scale-10; d’ = sensitivity index in the episodic memory task; K = capacity estimate in the short-term memory task.

## Data Availability

The data presented in this study are available on request from the corresponding author due to time limitations.

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
