# Peer review of "Nourishing the Brain or the Mood? Dietary Omega-3s for Psychological, but Not Cognitive Health"

_nutrients, 2025, doi:10.3390/nu18010050_

Round 1
Reviewer 1 Report
Comments and Suggestions for Authors
This study entitled Nourishing the Brain or the Mood? Dietary Omega-3s for 2 Psychological, but not Cognitive Health presents a methodologically sound investigation into the relationship between dietary omega-3 fatty acid intake and psychological versus cognitive health in healthy adults. It does not overstate claims and uses a sensible approach to analysing the data collected (i.e. PCA). However, the reviewer does note several areas that require further clarification:
- The sample is heavily skewed toward women who are highly educated, which may limit the generalisability of the findings. The authors should acknowledge this limitation more explicitly and discuss its implications for interpreting the results. The authors should also clearly state why there were a lack of men recruited (i.e. outline the barriers and why this occurred) and mostly educated people were recruited – the methods outlining participant recruitment require more detail and it is presumed that the only way to participate or a way of limiting participation of those less educated is due the fact the “massive” survey occurred purely online. Why were others not provided a means to complete this offline? This would provide a more heterogenous and diverse cohort leading to more data-rich study. Also, please remove the word “massive” and other non-academics words from the text.
- The study relies on self-reported dietary intake without biochemical validation, which introduces potential bias and limits the precision of omega-3 exposure estimates. Please outline specifically why blood or other clinical data could not be obtained even if this is due to cost. This should be noted as a limitation and then it should be stated that future studies should consider incorporating objective biomarkers. Additionally, the dietary analysis does not account for food preparation methods, which can significantly affect omega-3 content. Clarifying whether preparation techniques were considered would strengthen the dietary assessment. If this is not possible, please outline this is the limitations and again not it as a future direction.
- The moderation model didn’t show a significant effect, so the reviewer suggests shifting the focus away from that hypothesis. The stronger findings are in the mood and supplementation data, which are more consistent and meaningful. Reframing the paper in the discussion section to highlight those results would make the narrative clearer and better aligned with what the data actually support. Further, the finding that omega-3 supplement users had better episodic memory and fewer depressive symptoms is compelling. The authors should expand on the implications of this result, particularly in relation to dose consistency and bioavailability, which may differ from dietary sources which are all reported in the manuscript.
The English is fine, but there are some non-academic words used that are not suitable for publication with an example being "massive" to describe the number of participants completing the first phase.
Author Response
Answers to Reviewer 1 concerns
Comment 1: The sample is heavily skewed toward women who are highly educated, which may limit the generalisability of the findings. The authors should acknowledge this limitation more explicitly and discuss its implications for interpreting the results. The authors should also clearly state why there were a lack of men recruited (i.e. outline the barriers and why this occurred) and mostly educated people were recruited – the methods outlining participant recruitment require more detail and it is presumed that the only way to participate or a way of limiting participation of those less educated is due the fact the “massive” survey occurred purely online. Why were others not provided a means to complete this offline? This would provide a more heterogenous and diverse cohort leading to more data-rich study. Also, please remove the word “massive” and other non-academics words from the text.
Response 1:
We thank the Reviewer for this comment. We have now expanded the Limitations section to more explicitly address the demographic composition of our sample and its implications for generalisability (line 353). We clarified that the overrepresentation of women and highly educated individuals likely reflects a well-documented volunteer bias observed in online health-related studies, where women and individuals with higher education levels tend to show greater interest and engagement in lifestyle and mental health research.
We also added more detail in the Methods section (line 96) to clarify that participation in the first phase of the study was conducted entirely online through the Qualtrics platform. This approach was chosen for feasibility and to ensure standardized administration of the computerized cognitive tasks (hosted via Pavlovia.org). However, we acknowledge that this format may have unintentionally excluded individuals with lower digital literacy or limited internet access, thereby reducing the heterogeneity of the sample. We have also removed the term “massive” and other informal expressions.
Comment 2:
The study relies on self-reported dietary intake without biochemical validation, which introduces potential bias and limits the precision of omega-3 exposure estimates. Please outline specifically why blood or other clinical data could not be obtained even if this is due to cost. This should be noted as a limitation and then it should be stated that future studies should consider incorporating objective biomarkers. Additionally, the dietary analysis does not account for food preparation methods, which can significantly affect omega-3 content. Clarifying whether preparation techniques were considered would strengthen the dietary assessment. If this is not possible, please outline this is the limitations and again not it as a future direction.
Response 2:
We thank for this important comment. We agree that relying solely on self-reported dietary intake introduces potential bias and limits the precision of estimating omega-3 exposure. We have now explicitly acknowledged this limitation in the Discussion section and clarified that biochemical validation (e.g., plasma EPA and DHA levels) was not feasible in the present study in a large, primarily online sample. We have added a statement explaining that future research should incorporate objective biomarkers to strengthen the accuracy of omega-3 assessment.
Comment 3:
The moderation model didn’t show a significant effect, so the reviewer suggests shifting the focus away from that hypothesis. The stronger findings are in the mood and supplementation data, which are more consistent and meaningful. Reframing the paper in the discussion section to highlight those results would make the narrative clearer and better aligned with what the data actually support. Further, the finding that omega-3 supplement users had better episodic memory and fewer depressive symptoms is compelling. The authors should expand on the implications of this result, particularly in relation to dose consistency and bioavailability, which may differ from dietary sources which are all reported in the manuscript.
Response 3:
We thank the Reviewer for this constructive suggestion. We added two paragraphs to the end of the Results section to de-emphasize the moderation effect and highlight the more consistent findings. We also expanded the discussion of supplementation to address issues of dose stability, higher bioavailability of EPA/DHA in supplements relative to dietary sources, and the possible threshold effects required to influence cognition. We hope that these additions help better align the narrative with what the data support.
Reviewer 2 Report
Comments and Suggestions for Authors
The paper deals with the influence of diet, and in particular PUFA intake, on mood, mental state and cognitive health. The research was conducted correctly. Minor corrections are suggested as follows:
Abstract: Expressing the average as M is not common in the medical literature and is not necessary in the abstract. If you want to emphasize that, use mean (standard deviation).
Materials and methods: In the materials and methods, you say that you calculated the omega-3 DQI by generating a score from 0 to 100, and in Table 2 the score is not multiplied by 100. I suggest you correct Table 2.
Results: In the main text, you do not follow the SAMPL Guidelines, so instead of meand (SD), you write mean (SD = xy, range x -y). Please correct according to the guidelines. You say that the SD for BMI is 4.99, and in Table 2 you state that the max BMI was 61.73, which is neither mathematically nor biologically correct.
In paragraph 3.3 you refer to Table 1, and you should refer to Table 3. Below tables 3 - 8 you should state the name of the statistical test used, and not just an explanation of the abbreviations. In chapter 3.5 you forgot to refer to Table 5 in the main text, in chapter 3.5 you did not refer to Table 7, and in paragraph 3.6 to Table 8.
Check how statistical significance is written in the medical literature according to the SAMPL Guidelines.
Author Response
Comment 1: Abstract: Expressing the average as M is not common in the medical literature and is not necessary in the abstract. If you want to emphasize that, use mean (standard deviation).
response 1: Thank you for this suggestion. Following your recommendation, we have removed the abbreviation M entirely from the abstract.
Comment 2: Materials and methods: In the materials and methods, you say that you calculated the omega-3 DQI by generating a score from 0 to 100, and in Table 2 the score is not multiplied by 100. I suggest you correct Table 2.
response 2: Thank you for this comment. Table 2 correctly presents the factor loadings from the PCA, which were used to identify dietary patterns, and therefore does not display Omega-3 DQI scores. The Omega-3 DQI is a separate metric calculated on the raw consumption frequencies and ranges from 0 to 100, whereas Table 2 reflects the structure of the PCA components and not the DQI itself. For this reason, no multiplication by 100 is applicable in Table 2.
Comment 3: Results: In the main text, you do not follow the SAMPL Guidelines, so instead of meand (SD), you write mean (SD = xy, range x -y). Please correct according to the guidelines. You say that the SD for BMI is 4.99, and in Table 2 you state that the max BMI was 61.73, which is neither mathematically nor biologically correct.
Response 3: Thank you for this important observation. We have revised the entire manuscript to ensure full compliance with the SAMPL Guidelines, and all descriptive statistics are now reported in the standard format mean (SD) without additional expressions.
Regarding the BMI values, we acknowledge the inconsistency identified by the Reviewer. This error was located in Table 1 (not Table 2), and we have now corrected the maximum BMI value to reflect accurate and biologically plausible data. The corrected values are consistent across the text and tables. We appreciate the Reviewer bringing this to our attention.
Comment 4: In paragraph 3.3 you refer to Table 1, and you should refer to Table 3. Below tables 3 - 8 you should state the name of the statistical test used, and not just an explanation of the abbreviations. In chapter 3.5 you forgot to refer to Table 5 in the main text, in chapter 3.5 you did not refer to Table 7, and in paragraph 3.6 to Table 8.
Response 4: Thank you for these helpful remarks regarding clarity in table referencing. We have carefully reviewed the entire Results section and corrected all table citations.
Reviewer 3 Report
Comments and Suggestions for Authors
Introduction
Authors should include some references to specific populations, such as young people (a fragile subgroup but subject to adverse events from drug therapies):
Pruneti, C., & Guidotti, S. (2023). Need for Multidimensional and Multidisciplinary Management of Depressed Preadolescents and Adolescents: A Review of Randomized Controlled Trials on Oral Supplementations (Omega-3, Fish Oil, Vitamin D3). Nutrients, 15, 2306. https://doi.org/10.3390/nu15102306
Methods
Reliability values ​​for the psychological instruments are missing.
Furthermore, it is unclear whether the authors selected the direction of the relationships between the variables or whether they tested the fit of different models.
Discussions
It would be helpful to emphasize the difficulty of interpreting the directionality of the results. More specifically, depressed people generally eat worse and, consequently, reduce their intake of micronutrients important for their mental health, creating a vicious cycle in which depression is both a cause and a consequence. Perhaps it would be useful to consider diet as a mediating factor rather than a moderating one.
The authors should highlight the limitations of not directly controlling participants' food intake. For example, one study highlighted how analyzing diet and measuring subjects is crucial:
Guidotti, S., Fiduccia, A., Sanseverino, R., Pruneti, C. (2025). Multidimensional Assessment of Orthorexia Nervosa: A Case-Control Study Comparing Eating Behavior, Adherence to the Mediterranean Diet, Body Mass Index, Psychological Symptoms, and Autonomic Arousal. Nutrients, 17, 317. https://doi.org/10.3390/nu17020317
Author Response
comment 1:
Introduction
Authors should include some references to specific populations, such as young people (a fragile subgroup but subject to adverse events from drug therapies):
Pruneti, C., & Guidotti, S. (2023). Need for Multidimensional and Multidisciplinary Management of Depressed Preadolescents and Adolescents: A Review of Randomized Controlled Trials on Oral Supplementations (Omega-3, Fish Oil, Vitamin D3). Nutrients, 15, 2306. https://doi.org/10.3390/nu15102306
response 1:
Thank you for this suggestion. We have now incorporated a reference to research on younger and clinically vulnerable populations. Specifically, we added the study by Pruneti & Guidotti (2023) and described it in the discussion section.
comment 2:
Methods
Reliability values ​​for the psychological instruments are missing.
Furthermore, it is unclear whether the authors selected the direction of the relationships between the variables or whether they tested the fit of different models.
response 2:
Thank you for this comment. We have now added the reliability coefficients for all psychological instruments used in the study. In the Methods section, Cronbach’s alpha values are reported for both the PHQ-9 (α = 0.84) and the PSS-10 (α = 0.83), based on the present sample.
Regarding the direction and structure of the statistical model, we clarify that the analyses were theory-driven. The regression and moderation models were specified a priori based on established literature indicating that mood symptoms predict cognitive performance, with dietary omega-3 intake as a potential moderator. Therefore, we did not test alternative model structures or conduct bidirectional or exploratory model-fit comparisons. This clarification has now been added to the Methods for improved transparency.
Comment 3:
Discussions
It would be helpful to emphasize the difficulty of interpreting the directionality of the results. More specifically, depressed people generally eat worse and, consequently, reduce their intake of micronutrients important for their mental health, creating a vicious cycle in which depression is both a cause and a consequence. Perhaps it would be useful to consider diet as a mediating factor rather than a moderating one.
The authors should highlight the limitations of not directly controlling participants' food intake. For example, one study highlighted how analyzing diet and measuring subjects is crucial:
Guidotti, S., Fiduccia, A., Sanseverino, R., Pruneti, C. (2025). Multidimensional Assessment of Orthorexia Nervosa: A Case-Control Study Comparing Eating Behavior, Adherence to the Mediterranean Diet, Body Mass Index, Psychological Symptoms, and Autonomic Arousal. Nutrients, 17, 317. https://doi.org/10.3390/nu17020317
response 3:
Thank you for these insightful comments. We have now expanded the Discussion to more clearly address the issue of directionality and reverse causality. Specifically, we note that depressive symptoms may lead to poorer dietary habits, which in turn may further worsen mood, creating a bidirectional cycle. Additionally, we have strengthened the Limitations section by noting that we did not directly control or observe participants’ food intake.
Reviewer 4 Report
Comments and Suggestions for Authors
This is an interesting research study with quite adequate novelty which focuses on the effect of omega-3 fatty acids acids in psychological and cognitive health. However. some points should be added.
- In the results section of the abstract what (M = 40.30) and (M = 58.25) means?
- The authors should provide a breef report of the questionnaire used in the methods section of the abstract.
- In the 1rst paragraph, the authors should add some epidemiological data for mood disorders and cognitive impairment for their country.
- Please, briefly report some pro-inflammatory pathways whic are associed with depression risk (lines 47-48).
- A relevant reference should be added for the Mini Mental State Examinatiom (MMSE) questionnaire in lines 95-97.
- An explanaition for the cut off value of MMSE should be added.
- In the discussion section, the dosages of the omega-3 fatty acids used in the reported studies could be useful for the readers.
- In the limitations of the study, the used of online questionnaires should be reported as they may lead to recall bias.
- A conclusion section summarizing the most important results of the study, and reporting what future studies could be performed based on the results of the present study should be added.
Author Response
Comemnt 1:
In the results section of the abstract what (M = 40.30) and (M = 58.25) means? The authors should provide a breef report of the questionnaire used in the methods section of the abstract.
response 1: Thank you for this comment. To improve clarity, we have removed the statistical abbreviation “M” (mean) from the abstract and now report only the key descriptive values in plain language. We also added a brief description of the psychological instruments in the Methods section of the abstract, specifying that mood was assessed using the PHQ-9 and perceived stress using the PSS-10. These changes enhance readability and ensure that the measures are clearly introduced for readers.
Comment 2: In the 1rst paragraph, the authors should add some epidemiological data for mood disorders and cognitive impairment for their country. Please, briefly report some pro-inflammatory pathways whic are associed with depression risk (lines 47-48).
response 2:
Thank you for these suggestions. We have now expanded the Introduction to include epidemiological data specific to Poland. Additionally, we added a brief description of key pro-inflammatory pathways implicated in depression, including cytokine activation and immune–metabolic mechanisms. These additions provide clearer context for the relevance of dietary factors in mood and cognitive health.
Comment 3: A relevant reference should be added for the Mini Mental State Examinatiom (MMSE) questionnaire in lines 95-97. An explanaition for the cut off value of MMSE should be added.
Response 3:
Thank you for this suggestion. We have now added the standard reference for the Mini Mental State Examination (Folstein et al., 1975) in the Methods section. We also clarified that cut-off was set at 28 points to ensure inclusion of cognitively intact individuals..
Comment 4:
In the discussion section, the dosages of the omega-3 fatty acids used in the reported studies could be useful for the readers. In the limitations of the study, the used of online questionnaires should be reported as they may lead to recall bias. A conclusion section summarizing the most important results of the study, and reporting what future studies could be performed based on the results of the present study should be added.
Response 4: Thank you very much for these constructive suggestions. We have revised the manuscript accordingly and added a concise Conclusion section summarising the key findings of the study and outlining directions for future research.
Round 2
Reviewer 3 Report
Comments and Suggestions for Authors
I believe the authors have made a great effort to incorporate all of my proposed changes and suggestions.
However, I am skeptical about the originality of the work and its real impact on the clinical practice of healthcare professionals. I will leave the final decision to the editor.
Author Response
Thank you for your decision.
Reviewer 4 Report
Comments and Suggestions for Authors
The authors have significantly improved their manuscript.
Author Response
Thank you for your decision.